# 3D Active Brownian Motion of Single Dust Particles Induced by a Laser in a DC Glow Discharge

**DOI:** 10.3390/molecules28041790

**Published:** 2023-02-14

**Authors:** Anton S. Svetlov, Mikhail M. Vasiliev, Evgeniy A. Kononov, Oleg F. Petrov, Fedor M. Trukhachev

**Affiliations:** 1Joint Institute for High Temperatures, Russian Academy of Sciences, 125412 Moscow, Russia; 2Active Media and Systems Physics Laboratory, Moscow Institute of Physics and Technology, 141701 Dolgoprudny, Russia

**Keywords:** active particles, active Brownian motions, Janus particle

## Abstract

The active Brownian motion of single dust particles of various types in the 3D electrostatic DC discharge trap under the action of laser radiation is studied experimentally. Spherical dust particles with a homogeneous surface, as well as Janus particles, are used in the experiment. The properties of the active Brownian motion of all types of dust particles are studied. In particular, the 3D analysis of trajectories of microparticles is carried out, well as an analysis of their root mean square displacement. The mean kinetic energy of motion of the dust particle of various types in a 3D trap is determined for different laser powers. Differences in the character of active Brownian motion in electrostatic traps with different spatial dimensions are found.

## 1. Introduction

Brownian motion is the random motion of particle suspended in a liquid or a gas [1]. This pattern of motion describes an environment at thermal equilibrium, defined by a given temperature. There is no preferred direction of movement of Brownian particles. Brownian motion is named after the botanist Robert Brown, who first described the phenomenon in 1827. “While he was studying microscopic life, he noticed little particles of plant pollens jiggling around in the liquid he was looking at in the microscope, and he was wise enough to realize that these were not living, but were just little pieces of dirt moving around in the water” [1]. The first scientific consideration of Brownian motion was given by Albert Einstein in 1905 [2]. Einstein associated the observed phenomenon with the constant random bombardment of a Brownian microparticle by the molecules (atoms) of the liquid. Jean Perrin was awarded the Nobel Prize in 1908 for his experimental study of the properties of Brownian motion.

It is important to note that classical Brownian particles are in thermodynamic equilibrium with close ones, in contrast to active Brownian particles. Particles that can convert the energy of the environment into the energy of their own directed motion are called active Brownian particles (or self-propelled Brownian particles) [3,4]. Self-consistent motion of active particles is usually called active motion. Examples of active particles include some bacteria, mobile cells [5,6,7], micro- and nanorobots [8], active microcatalysts [9,10,11], microdroplets in an emulsion [12,13], dust particles in discharge plasma and superfluid helium [14,15,16,17,18] etc. Active particles can either move independently or demonstrate a collective nature [5,19]. The mean kinetic energy of active particles can significantly exceed the mean kinetic energy (temperature) of the environment, which indicates a significant nonequilibrium of the process [3]. The main mechanisms of activity of microparticles are: chemical reactions [9,10,11], Marangoni effect [12], quantum effects [18], ion drag forces and photophoretic forces in colloidal plasma [14,15,16,17], external electromagnetic fields [13], etc.

Initially, the applied significance of the active Brownian particles physics consisted in the development of methods for increasing the catalysis efficiency. Over time, the scope of applications has expanded significantly and currently includes: research in the field of drug delivery, new ecologic technologies, development of new materials, etc. The fundamental aspect of research is the development of the physics of nonequilibrium systems, physics of dissipative structures, chaos physics, principles of self-organization, self-assembly, and the origin of life.

Colloidal plasma is a convenient object for studying active Brownian particles [15]. On the one hand, in plasma, compared to electrolytes, chemical reactions play a much smaller role, which makes it possible to more accurately control the process parameters. On the other hand, the significantly lower plasma viscosity makes it possible to use weak sources of active motion energy. The results of an experimental study of the active Brownian motion of single dust particles under the action of laser radiation in a RF discharge plasma were presented in [15]. Dust particles with a uniform coating demonstrated mainly orbital motion in the symmetric electrostatic trap of a RF discharge, their trajectories were close to circles of different diameters depending on the laser power. At the same time, the dynamics of dust Janus particles was a combination of orbital and random motions. Janus particles are types of microparticles whose surfaces have two (or more) different physical properties [20]. According to [15], the main reason for the active Brownian motion of all types of dust particles was the photophoretic force. The experiments presented in [15] had a two-dimensional (2D) topology, since they were carried out in an RF discharge in a monolayer configuration. 

In our work, we present studies of the active Brownian motion of the single dust particles in the three-dimensional (3D) configuration of a DC glow discharge. The aim of this work is to compare the dynamics of dust particles with different surface properties in the same plasma environment. Our work contains an analysis of the trajectories of the dust particles, as well as an analysis of their root mean square (RMS) displacement. The calculation of the mean kinetic energy of the dust particles of various types under the action of laser radiation has been carried out. These methods of analysis are among the main ones in the study of active Brownian motion [4,10,14]. 

## 2. Data Analysis and Discussion

### 2.1. Materials

The following types of spherical melamin-formaldehyde (MF) dust particles with a diameter of d ≈ 5 µm are used in the experiment: uncoated dust particles, dust particles with a uniform thin copper coating (Cu-coated), and particles with partial copper coating (Janus particles). Photoimages of the dust particles obtained using a scanning electron microscope (SEM) are shown in Figure 1. Note that in our case, dust particles of half the size compared to the experiments [15,18] are used. This is due to the electrostatic configuration of the DC discharge, which does not allow for the levitation of heavy dust particles under gravitational conditions. Uncoated and Cu-coated dust particles were produced by MicroParticles GmbH (Berlin, Germany), while Janus particles were manufactured in our laboratory according to the technology described in [21]. 

Let us here briefly describe the method of Janus particles manufacturing. The process was based on the modification of the surface of microparticles under the action of ion flows in the discharge plasma. The manufacturing was carried out inside the capacitive RF discharge chamber. Spherical uncoated MF dust particles (d ≈ 5 µm) were placed on the bottom electrode, where they were held by the gravity force. The surface density of the particles at the bottom electrode was about 2000 cm^−2^. After that, the discharge chamber was evacuated (to a residual pressure *p_Air_* < 10^−2^ Pa) and filled with an inert buffer gas (argon), to a working pressure *p_Ar_* = 5 Pa. During the modification process, the discharge chamber was continuously evacuated by a turbomolecular pump, the pressure was maintained by a continuous supply of working gas at a rate of 2 standard cm^3^/min. Thus, the discharge plasma retained unchanged properties throughout the entire Janus particle manufacturing process. To ignite the RF discharge, an AC voltage was applied to the electrodes from a high-frequency generator with a frequency of 13.56 MHz. The plasma was generated at a power *W_load_* = 15 W, while the reflected power was *W_ref_* = 4 W. During the existence of a discharge with the considered parameters, sputtering of copper electrodes was observed, which led to the appearance of copper ions in the plasma volume. In the near-electrode layer (plasma sheath), plasma ions were accelerated to supersonic speeds *υ_i_* > *C_s_* (*C_s_* denotes the ion-sound speed) and bombarded the surface of the processed microparticles. It is important to note that copper ions formed a thin copper coating on the surface of the microparticles, while argon ions eroded their surface. The processes under consideration took place in the region of the upper hemisphere of microparticles (facing towards the discharge), which is associated with the fulfillment of the condition *υ_i_* > *C_s_*. As a result, the surface of the microparticles was modified only on one side. At the same time, the surface of the microparticles on the opposite side retained its original properties. Thus, initially homogeneous dust particles acquired the properties of Jancus dust particles. The optimal exposure time for microparticles was experimentally determined, which was approximately 5 h for the considered discharge parameters. At lower exposures, the copper coating was fragmented, while at longer exposures, strong erosion of the irradiated hemisphere was observed (see [21]). The appearance of copper on the surface of modified MF particles was confirmed by X-ray spectral analysis. The mass fraction of copper was a few percent. In addition, the surface modification was confirmed by electron microscopy (SEM) image analysis.

### 2.2. Analysis of Dust Particle Trajectories

The dynamics of a single dust particle in a DC discharge is determined by a number of forces, the main of which are: gravitational force, *F_g_*; electric force of the constant discharge electric field, *F_E_*; ion and neutral drag forces *F_id_*, *F_nd_;* photophoretic forces *F_ph_* [22]. Often, the first three forces *F_g_*, *F_E_*, *F_id_* provide the state of equilibrium for levitating dust particles. The dust particles under studies had approximately the same mass and size (according to our estimates, the difference in dust particle masses is ~35%, due to the mass of a thin copper shell). Therefore, with the same discharge parameters, we have *F_g_*_1_ ≈ *F_g_*_2_ ≈ *F_g_*_3_ and *F_id_*_1_ ≈ *F_id_*_2_ ≈ *F_id_*_3_, where the subscripts “1”, “2”, “3” denote different types of the dust particles (i.e., uncoated, Cu-coated and Janus particle). The electrical force *F_E_* is determined by the charge of dust particles and electric field strength of the DC discharge, *E*. For isolated dust particles, the orbital motion limited approximation (OML) can be used. According to known models [23], the charge of a dust particle immersed in a gas discharge depends on the surface area. Therefore, the particle charge can be considered the same, then *F_E_*_1_ ≈ *F_E_*_2_ ≈ *F_E_*_3_. Thus, the dynamics of all three types of dust particles in the state of equilibrium (in the absence of laser radiation) is approximately the same. In this case, it is not necessary to accurately measure the above forces, but it is sufficient to ensure the same discharge conditions for all types of dust particles. Let us consider the forces that determine their active Brownian motion. It was shown in [15] that the activity mechanisms that bring the 2D plasma-dust system out of equilibrium under the action of laser radiation are based on the following phenomena:
Photophoresis due to a temperature gradient on the dust particle surface, that is an asymmetric neutral drag force caused by a temperature difference (for all dust particles); Photophoresis due to different accommodation coefficients, that is a neutral drag force caused by different accommodation coefficients of MF and copper (for Janus particles).


The photophoretic force is the force associated with the light radiation. Temperature gradients, along with the rotation of a dust particle around an axis, can create forces that cause its active Brownian motion [24]. Obviously, the action of the photophoretic force is significantly different for dust particles with different surface types [15]. A dust Janus particle illuminated by strong laser radiation can have the highest temperature gradients on the surface due to its inhomogeneity. This results in an intense photophoretic motion that can be controlled by adjusting the laser intensity, *I_las_*, which is proportional to the laser power *W_las_*~*I_las_*. Note that for dust particles with a uniform surface, photophoresis is also the cause of activity.

To study the active Brownian motion of single dust particles with different types of surfaces in the 3D configuration, the single microparticle is injected into the plasma volume of the DC discharge, where, as a result of the balance of the forces *F_g_*, *F_E_*, and *F_id_*, their levitation is observed. To excite the active motion, the levitating particles are irradiated with laser beam of various intensities. To analyze the 3D dynamic process in our experiment, we used two synchronized video cameras that record the movement of the dust particles in two perpendicular planes, which made it possible to determine the particle coordinates in space at any time. The results of video data processing are shown in Figure 2 and Figure 3.

As can be seen, an increase in the amplitude of active Brownian motion with increasing laser radiation intensity is a common feature of the dust particles of all types. At the same time, the movement parameters varies from one dust particle type to another. Uncoated dust particles make motions of the smallest amplitude. Their trajectories are extended along the laser beam direction. As the value of *W_las_* increases, the trajectories of uncoated dust particles become almost circular in the horizontal plane. The 3D nature of the particle dynamics is seen from the nonzero vertical component of the active Brownian motion (in contrast to [15]). Movement in the vertical direction is random, and its amplitude weakly depends on the laser power *W_las_*.

Cu-coated dust particles absorb laser radiation much more efficiently than uncoated dust particles. Consequently, the efficiency of the photophoretic phenomena increases [25]. This, in turn, leads to a noticeable increase in the amplitude of active Brownian motion in the horizontal plane. Horizontal trajectories of the Cu-coated dust particles have a shape close to ellipsoidal, elongated along the direction of the laser beam (see Figure 2). As in the previous case, the vertical component of the active Brownian motion of Cu-coated dust particles is relatively small and practically does not depend on the laser power *W_las_* (see Figure 3). 

The active Brownian motion of the Janus particles is the most complex. The extension of their trajectory strongly depends on the laser power *W_las_*. Practically no regular (elliptical and circular) tracks are observed, although some sections of the trajectories had the shape of arcs at high laser intensity. Inflections connecting adjacent arcs are often observed. The amplitude of the active Brownian motion increases with increasing laser power *W_las_*, both horizontally and vertically.

Theoretical model [26] satisfactorily describes the active Brownian motion of coated dust particles and Janus particles. Indeed, the motion of Cu-coated dust particles corresponds to “noise-free motion mode of a circle swimmer with constant self-propulsion in a constant spatial trap” (see Figure 2a,b from [26]). At the same time “noise-free motion mode of a circle swimmer with temporally varying self-propulsion force in a constant spatial trap” (Figure 3 from [26]) partially corresponds to the Janus particles active motion. It is important to note that experimental trajectories of Janus particles differ from the theoretical ones [26]. In both cases, the particle trajectories consist of arcs, but the connection between the arcs in the experiment is more stochastic than in theory.

Our results are in a good agreement with the results obtained in a quasi-two-dimensional electrostatic trap of an RF discharge [15]. At the same time, the active Brownian motion in our experiments is less regular than in [15] and exhibits a 3D nature. In our opinion, these differences are associated with different geometries of traps in RF and DC discharges.

Using an automated video image processing system, the time dependences of the root mean square (RMS) dust particle displacement are calculated, which are determined by the well-known formula [27]:<***r***^2^(*t*)>=<[***r***(*t*) − ***r***(0)]^2^>(1)
where ***r***(0) is the vector of the dust particle initial position, ***r***(*t*) is the position of the dust particle at time *t*. In the case of a single dust particle, time averaging is performed, the corresponding graphs are shown in Figure 4.

As can be seen, the nature of the active Brownian motion is similar to the motion of single dust particles in a 2D trap considered in [15]. The time scales in our case are about an order of magnitude shorter. This is explained by the fact that our dust particles are twice as small as the dust particles from the experiments described in [15]. In the cases under consideration, their masses differ by an order of magnitude. On a short time scale, the graphs show the ballistic motion regime with the asymptotic ~*t*^2^ (green line) for all types of dust particles. Starting from the time *t* ~ 0.1 s, the “trapped” motion mode with a certain period is observed (see also [28], Figure 4b at a = 1). As in [15], there is no asymptotic of ~*t*^3/2^ typical for active Brownian motion in classical studies for extended structures without any traps [4].

Figure 5 shows the dependence of the mean kinetic energy of the active Brownian motion of the single dust particles *T_k_* on the laser power *W_las_*. The averaging time is 60 s. The kinetic energy of the considered dust particles increased with increasing laser power in the range from 300 to 1300 mW as follows: *T_k_* ≈ 1 eV to *T_k_* ≈ 14 eV for uncoated dust particle; from *T_k_* ≈ 3 eV to *T_k_* ≈ 270 eV for Cu-coated dust particle; from *T_k_* ≈ 14 eV to *T_k_* ≈ 340 eV for Janus particle.

As can be seen, the kinetic energy of uncoated dust particles is low in the entire range of laser power, since they weakly absorb laser radiation (see Figure 5a). The same scenario was observed in [15]. For a coated dust particle, as well as for Janus particles, a significant increase in kinetic energy can be explained by an increase in the photophoretic force due to the effective light absorption by the dust particle surface. There is a “jump” for coated dust particle and dust Janus particle on the plot of *T_k_*(*W_las_*). To clarify the reasons for this phenomenon, we calculated the mean kinetic energy of dust particle active Brownian motion along the tube axis, $T_{k∥}=0.5\;m_{d}<v_{z}^{2}>$ and across it $T_{k⊥}=0.5\;m_{d}<v_{x}^{2}+v_{y}^{2}>$, where $m_{d}$ is the dust particle mass, $v_{x}^{},\;v_{y}^{}v_{z}^{}$ are dust particle velocity components. The corresponding plots are shown in Figure 5b,c. As can be seen, only the dependence $T_{k∥}\left(W_{las}\right)$ has a distinct “jump”. The strong axial inhomogeneity of the electric field in the striations of the DC discharge [29] is a possible reason for the observed phenomenon. 

## 3. Experimental Setup

Figure 6 shows the scheme of the experimental setup. The experiment was carried out in a vertically oriented glass discharge tube with a length *l* = 1250 mm and an inner diameter *D* = 40 mm. The distance between the cathode and anode was 1050 mm. The lower end of the tube was closed while the upper end was connected to a pressure control system. In addition, on top was the injector of dust particles. The tube was evacuated using a turbomolecular pump and filled with an inert gas (argon) to a working pressure of *p* = 10 Pa. Next, using a “Spellman” power source, a DC glow discharge was ignited between the anode and cathode. The main discharge parameters are as follows: voltage *U* = 2.0 kV and current *I* = 1.25 mA. During the experiment, argon was continuously fed into the discharge tube at a rate of 4 standard cm^3^/min, which made it possible to maintain the properties of the plasma unchanged.

Dust particles were injected into the discharge chamber by magnetic action (shaking) of the container with dust particles. After the injection, the dust particles fell into the region of the positive column of the discharge, where they were charged and captured in the ionization regions (stratum). To stabilize the position of the striations, a dielectric cone-shaped insert was used, which concentrated the flow of electrons and ions along the axis of the tube.

To visualize and influence the particles, we illuminated them with a uniform beam of an argon laser with a wavelength of 488 nm. Using an optical system of lenses, the laser beam was expanded to a radius of 10 mm and directed in such a way that it completely covered the region of particle motion and was perpendicular to the axis of the glass tube.

The movement of dust particles was recorded by two high-speed video cameras located at an angle of 90 degrees. The location was chosen so that the lenses are at the same height. At the same time, the axes were objectively located perpendicular to the axis of the discharge tube of the tube. The laser beam did not fall into the lenses of the video cameras and did not interfere with the registration process. The recording frequency used is 400 fps and a resolution of 1440 × 1440 (or 19.6 microns/pixel). These parameters made it possible to obtain high-quality and detailed video images. Each video camera creates a two-dimensional image of the observed phenomenon. To visualize the three-dimensional dynamics of dust particles, the images of both cameras were synchronized and processed by special software. The array of data obtained in this way made it possible to analyze the trajectories of single dust particles, as well as their velocities, and hence the kinetic energies.

## 4. Conclusions

The nature of the motion of single spherical MF dust particles with different surface types in the volume of the lower stratum of the DC glow discharge under the action of laser radiation has been experimentally studied. All considered particles had approximately the same size (*d* ≈ 5 µm) and weight. All differences are in the thin surface layer. In particular, uncoated dust particles, Cu-coated dust particles, and partially Cu-coated dust particles (Janus particles) are analyzed. The experiments are carried out at the same discharge parameters for all considered types of dust particles. 

For 3D analysis, the recording technique was used with two synchronized video cameras. The camera lenses were placed at an angle of 90°. The resulting images were processed on a computer using special software to obtain 3D dynamic characteristics of the studied dust particles.

It is shown that dust particles in the three dimensional DC discharge electrostatic trap perform the active Brownian motion under the action of laser radiation, i.e., convert radiation energy into kinetic energy of their own regular motion. The reason for the active Brownian motion, in our opinion, is the phenomenon of photophoresis.

It has been experimentally shown that the effect of laser radiation on an uncoated dust particle is the smallest in comparison with other types of particles. At the same time, for the Cu-coated dust particles and Janus particles, an increase in the laser power led to a significant increase in their kinetic energy and to an expansion of their region of motion. The trajectories of dust particles with a uniform surface are a superposition of ellipses extended along the laser beam direction with a random component. The active Brownian motion of the Janus particles can be characterized by the least regularity and the greatest three-dimensionality.

Thus, dusty Janus particles exhibit the most pronounced properties of active Brownian motion. It should be expected that a large ensemble of Janus particles will acquire a new physical property, consisting in a laser-controlled diffusion coefficient. On the other hand, it will be possible to control the kinetic temperature of the dust component by low-intensity laser radiation. It is important that the other plasma parameters (pressure, temperature, density, etc.) remain almost constant in this case. In turn, the control of the kinetic temperature of the dust component is associated with the control of the coupling parameter *Γ*, the dust-acoustic velocity, the pressure of the dust component, etc.

## Figures and Tables

**Figure 1 molecules-28-01790-f001:**
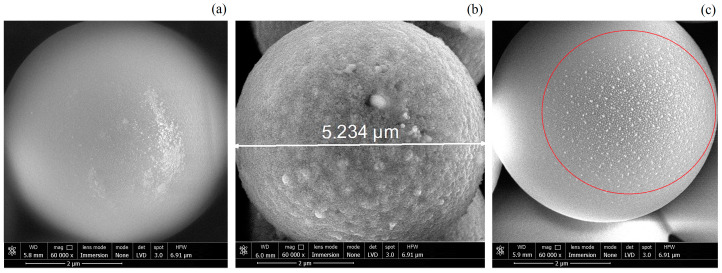
SEM photographs of the spherical melamine-formaldehyde (MF) dust particles with a diameter of 5 μm: (**a**) uncoated; (**b**) Cu-coated; (**c**) Janus particle.

**Figure 2 molecules-28-01790-f002:**
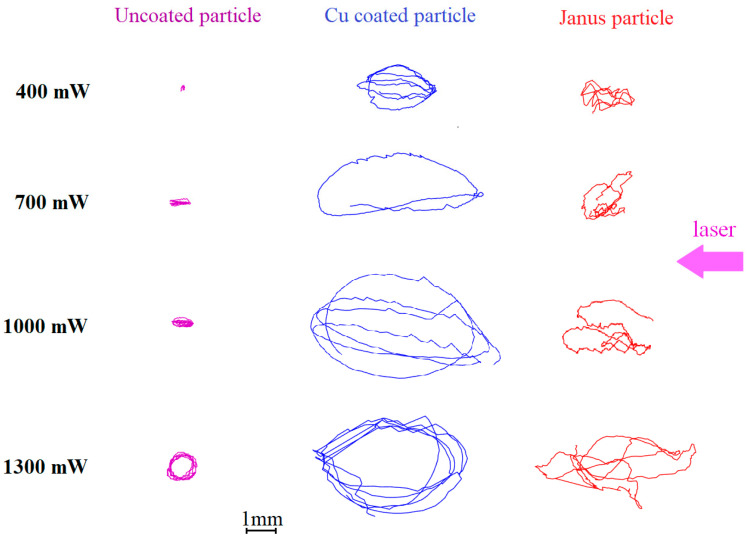
Top View (horizontal component of the active Brownian motion). Dust particle trajectories in the DC discharge stratum during the time Δ*t* = 1 s, at different laser powers *W_las_*: 400, 700, 1000 and 1300 mW. Left row is uncoated dust particle; the central row is the Cu-coated dust particle; right row is the dust Janus particle.

**Figure 3 molecules-28-01790-f003:**
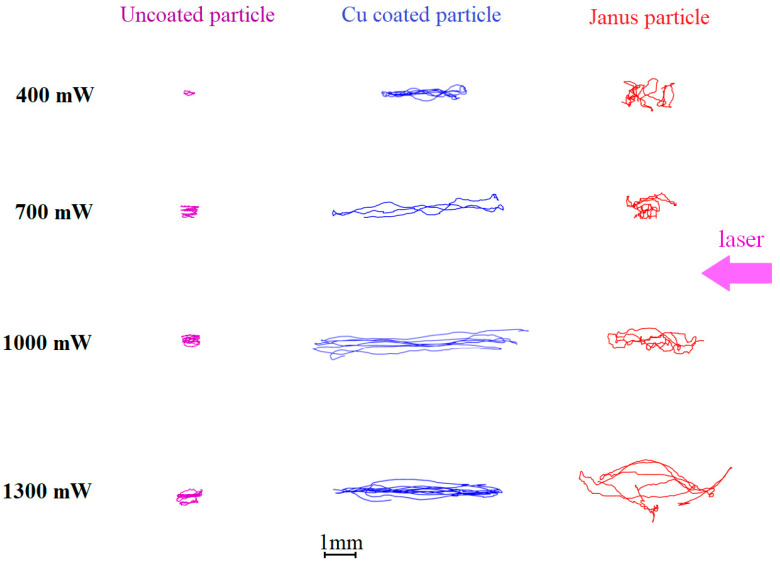
Side View (vertical component of the active Brownian motion). Dust particle trajectories in the DC discharge stratum during the time Δ*t* = 1 s, at different laser powers *W_las_*: 400, 700, 1000 and 1300 mW. Left row is uncoated dust particle; the central row is the Cu-coated dust particle; right row is the dust Janus particle.

**Figure 4 molecules-28-01790-f004:**
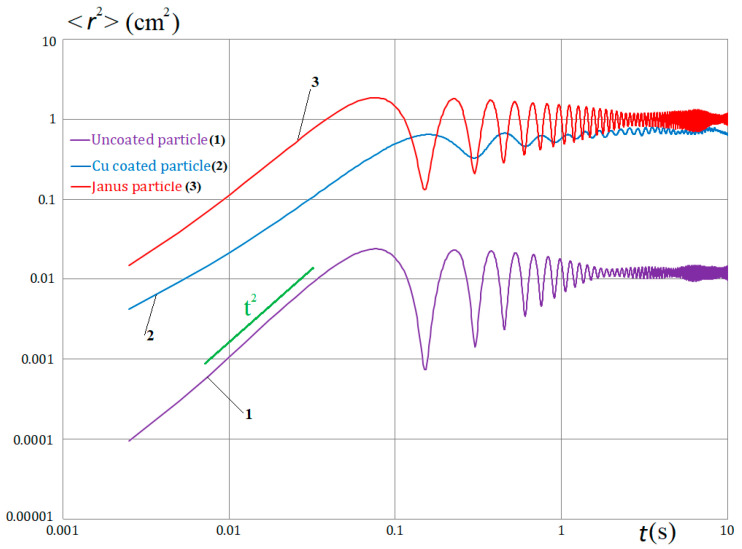
Root mean square (RMS) displacement of the single dust particles under the action of a 1300 mW laser: curve 1 for uncoated dust particle; curve 2 for Cu-coated dust particle; curve 3 for Janus particle.

**Figure 5 molecules-28-01790-f005:**
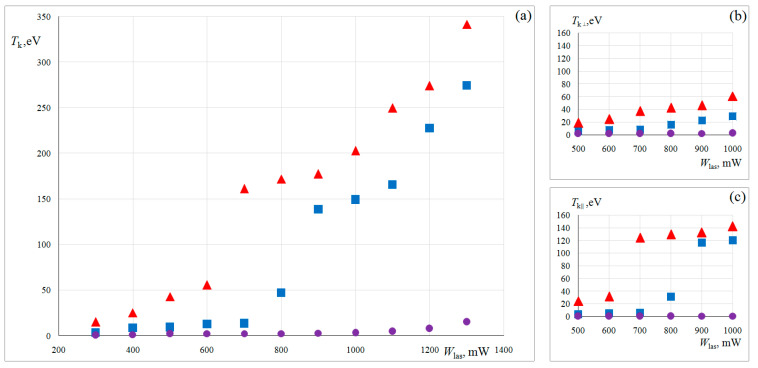
(**a**) Dependence of the mean kinetic energy of single dust particles on the laser power *W_las_*; (**b**) the same, but for the longitudinal (axial) component of the active Brownian motion; (**c**) the same, but for the transverse (radial) component of the active Brownian motion. Uncoated dust particle (circles); Cu-coated dust particle (squares); Janus particle (triangles).

**Figure 6 molecules-28-01790-f006:**
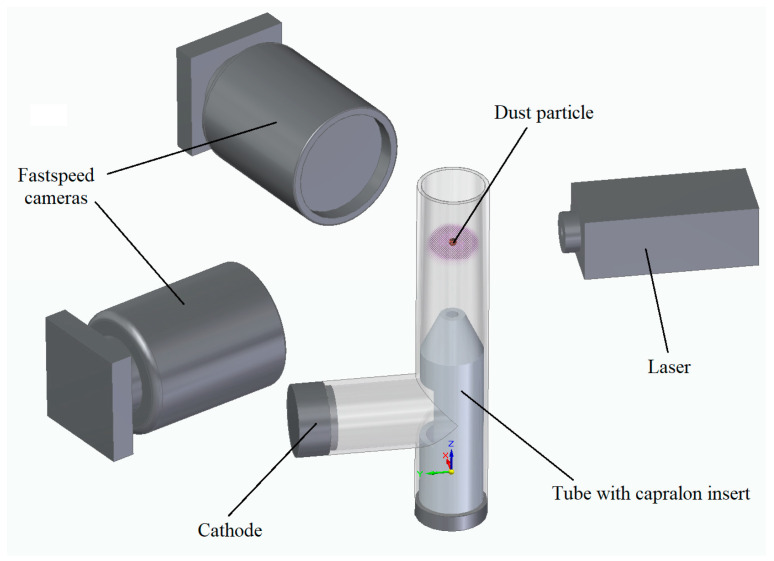
Scheme of the experimental setup.

## Data Availability

Data is contained within the article.

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
