# Peer review of "3D Active Brownian Motion of Single Dust Particles Induced by a Laser in a DC Glow Discharge"

_molecules, 2023, doi:10.3390/molecules28041790_

Round 1

Reviewer 1 Report

Review of :  Article

Title: 3D active Brownian motion of single dust particles induced by a laser in a DC glow discharge

By:  Svetlov et al

Journal: Molecules

I am not too enthusiastic about this paper.  It has some fairly interesting experimental results, but is very short on analytical content, including of some results from the experiments that are apparent upon brief inspection. 

The paper is about "active Brownian motion" of 3 types of particles, induced by a laser.  "Active Brownian motion" is well-described in the first two paragraphs.  It basically is "directed motion" of particles under the input of energy from some internal or external (in the case of the laser) source of energy.  Whereas otherwise one would expect completely random Brownian motion of the type famously explained by Einstein.  One can say that rectified Brownian motion is responsible for the "purposeful" behavior of things like biological motors, at least on a small enough time scale.  One might say that the rectification is a kind of symmetry breaking enabled by an input of free energy. 

In this paper the free energy input comes from a laser.  One apparently sees three types of non-random motion of three types of particles, as shown in Fig. 2 which apparently shows the "active component" of the motion – it is not too clear to me exactly what this means.  But the three types of particles show very different motions, though all seem to have a circular component as well as something random.  The Janus in particular has quite a lot of both.  Fig. 3 shows the same motions from a side view.  Fig. 4 seems to show some sort of averaging over many trajectories, plotted for a long time scale.  The Janus and coated particles clearly have larger amplitude of motion than the uncoated particles. 

What is lacking in this paper is any kind of analysis beyond the simple description, which I have attempted to summarize above.  Why do the coated particles so much larger amplitude motion than the uncoated?  Why the pronounced differences between the large amplitude coated and Janus particle motions? 

In Fig. 5 a graph of energy of the particles as a function of laser power.  The coated and uncoated each have a pronounced "jump" in the midrange of laser power.  Surely this means something or there is a reason for it?  But I can't find anything from the authors about this striking observation! 

Maybe other reviewers will think that there is enough here for a paper, but for my taste, it is kind of thin! 

Author Response

We thank the Referee for raising a number of important questions.  Our response to the issues in the attachment.

Reviewer 2 Report

In this study, The authors have experimentally investigated the active Brownian motion of various types of individual particles in 3D electrostatic DC discharge traps under the influence of laser radiation, using spherical particles with uniform surfaces and Janus particles. The discussions in the paper are well-founded, and all calculations are well-presented. However, the manuscript involves some minor errors that should be corrected before publication:

  1. In order to better understand the latest research progress for active Brownian particle physics, the author should add the latest literature in theintroduction.
  2. The preparation process of Janus particles should be added to this manuscript.
  3. Page 5, line 157, “Tk≈3 eV” should be revised to “Tk≈3 eV”.
  4. In Page 5, the author described the dependence of the mean kinetic energy of motion of single particles Tk on the laser power, but there was no explanation for the discrepancy of this part.
  5. In Figure 2, the properties of Janus particles are different between 700 mW and 1000 mW. Please explain the reason for this result.
  6. In the article, line 131 mentions that “In our opinion, these differences are associated with different geometries of traps in RF and DC discharges.” what is the basis for this conclusion?
  7. The fifth and sixth data of the Janus particle in Figure 5 are clearly above the standard, please explain it.
  8. Please use the same format between symbols and numbers/letters.
  9. Please unify the format of references.

Author Response

(The authors gave the same response as above.)

Reviewer 3 Report

The paper from Anton S. Svetlov et al. is devoted to the investigation of  the Brownian motion of particles of various types, induced by laser radiation. The article is a logical continuation of the previously published work [Arkar, K.; Vasiliev, M.M.; Petrov, O.F.; Kononov, E.A.; Trukhachev, F.M. Dynamics of Active Brownian Particles in Plasma. Molecules 2021, 26, 561] in terms of the transition from the study of particle motion in a plane to three-dimensional motion. Methodically, the article is quite close to the one mentioned above.

In general, the article is quite interesting, but there are shortcomings in it, the correction of which could improve the quality of the article.

1. Probably, the authors erroneously indicated that the particles are composed of melanin-formaldehyde: "The following types of spherical melanin-formaldehyde (MF) particles ...". Apparently, melamine-formaldehyde was meant.

 2. On page 2, the authors write: "Photoimages of the particles obtained using a scanning electron microscope (SEM) with a spatial resolution of 0.4 nm are shown in Figure 1." Figure 1 shows an image of particles with a diameter of 5 µm. At such an image scale, details of the order of 0.4 nm in size are simply not visible (the size of the area corresponding to one pixel in the figure is slightly larger). So in this case, specifying such a resolution is not entirely correct.

 3. Figures 2 and 3 show the motion trajectories for individual particles of different types. Moreover, for each type of particles and a certain radiation power, only one trajectory is given. In my opinion, it would make sense to present several trajectories for at least one case, which would demonstrate the reproducibility of the results obtained. Perhaps this will require the addition of a separate figure in the article. 

4. On page 2, there is a statement: "The particles under studies had approximately the same mass and size." In conclusion, this statement becomes even more strict "All particles had the same size (D≈5 μm) and weight". However, even based on the images shown in Fig. 2, it can be seen that the particles size is somewhat different. In addition, it is obvious that the deposition of even a small layer of copper on the surface of a particle can significantly increase its mass. From Figure 2(b), it is obvious that the copper layer on the surface of the particle is thick enough. So I would recommend the authors to give at least some characteristics of the distribution of particles by mass and size.

After correcting these shortcomings, the article can be accepted for publication

Author Response

(The authors gave the same response as above.)
